# The Outcomes of COVID-19 Patients with Spontaneous Intracerebral Hemorrhage Comorbidity and the Efficacy of Enoxaparin in Decreasing the Mortality Rate in Them: Single Egyptian Center Report

**DOI:** 10.3390/jpm12111822

**Published:** 2022-11-02

**Authors:** Mohamed Shaban, Marwa O. Elgendy, Alzhraa M. Fahmy, Doaa Mahmoud Khalil, Ahmed O. El-Gendy, Tamer M. Mahmoud, Mohamed E. A. Abdelrahim

**Affiliations:** 1Neurosurgery Department, Faculty of Medicine, Beni-Suef University, Beni-Suef 62521, Egypt; 2Department of Clinical Pharmacy, Teaching Hospitals of Faculty of Medicine, Faculty of Medicine, Beni-Suef University, Beni-Suef 62521, Egypt; 3Department of Clinical Pharmacy, Faculty of Pharmacy, Nahda University (NUB), Beni-Suef 62764, Egypt; 4Tropical Medicine and Infectious Diseases Department, Faculty of Medicine, Beni-Suef University, Beni-Suef 62521, Egypt; 5Public Health and Community Medicine Department, Faculty of Medicine, Beni-Suef University, Beni-Suef 62521, Egypt; 6Department of Microbiology and Immunology, Faculty of Pharmacy, Beni-Suef University, Beni-Suef 62521, Egypt; 7Internal Medicine Department, Faculty of Medicine, Beni-Suef University, Beni-Suef 62521, Egypt; 8Clinical Pharmacy Department, Faculty of Pharmacy, Beni-Suef University, Beni-Suef 62521, Egypt

**Keywords:** neurological disorders, enoxaparin, spontaneous intracerebral hemorrhage, COVID-19

## Abstract

Patients with neurological comorbidities are more likely to develop severe COVID-19. We aimed to detect the outcomes of COVID-19 patients with spontaneous intracerebral hemorrhage comorbidity and the role of enoxaparin in decreasing the mortality rate in these cases, even though enoxaparin is a potential cause of intracerebral hemorrhage. The patients were checked on to detect surveillance outcomes, the relationship between mortality and patient characteristics, and the relationship between enoxaparin and study outcomes. Chest condition and GCS improved in 67.9% of participants. Hematoma course increased in 49.1%. Midline-shift, brain-edema, and COVID symptoms improved in 67.9%. There was a non-significant difference in mortality regarding age and gender. There was a significant difference in mortality regarding treatment with enoxaparin; 75% of the patients who did not receive enoxaparin died. 92.6% of the patients who showed decreases in hematoma course were administered enoxaparin. 76.9% of the patients who showed increases in hematoma-course were administered enoxaparin. Most of the patients who were admitted to the neurosurgical unit with spontaneous intracerebral hemorrhage acquired the COVID-19 infection. Most of the cases included in this study did not progress to severe cases. The dying patients showed deterioration in both neurological and COVID-19 symptoms. The anticoagulant properties of enoxaparin given earlier before and throughout the infection can considerably reduce mortality in COVID-19 individuals with spontaneous intracerebral hemorrhage. It is recommended to use enoxaparin for cases with spontaneous intracerebral hemorrhage and COVID-19 regardless of hematoma size because the rate of improvement was greater than the mortality rate after using enoxaparin in this study.

## 1. Introduction

Coronavirus infection is caused by the severe acute respiratory syndrome coronavirus 2 (SARS-CoV-2) [1]. It often appears as respiratory tract symptoms and fever [2,3,4,5]. To evaluate the patients with COVID-19 infection, the Dutch Radiological Society developed a score system based on chest CT and patient data in March 2020; the COVID-19 Reporting and Data System (CO-RADS) included clinical findings and laboratory test results, as well as CT records. The level of suspicion ranged from very low to very high (CO-RADS categories 1–5), with category 0 indicating no infection and category 6 indicating RT-PCR-positive SARS-CoV-2 infection at the time of examination. [1]. In clinical settings, coronavirus may cause severe symptoms, particularly in elderly patients with many comorbidities [6]. The risk factors (comorbidities) for severe coronavirus symptoms include cerebrovascular disease, diabetes mellitus, cardiovascular disease, hypertension, and intracerebral hemorrhage (ICH) [7]. A growing number of studies on coronavirus have reported many neurological complications, such as Guillain–Barré syndrome [8]., acute stroke, hyposmia, and encephalitis [9,10]. It has been demonstrated that approximately 30% of coronavirus patients develop neurological complications, which are frequently linked to a more severe infection, implying that coronavirus neurotropism is a possible mechanism of neurological damage [11,12]. ICH is indeed a very rare but well-documented COVID-19 complication [13]. Therefore, ICH can be regarded as a risk factor (comorbidity) and a complication of severe coronavirus infection. Many research studies looked at the ICH as a COVID-19 complication, but few published studies look at the ICH as a pre-existing comorbidity for severe coronavirus infection. Accordingly, we decided to conduct an observational study to detect the outcomes of COVID-19 patients with pre-existing spontaneous ICH (comorbidity). Primary ICH can happen in the presence or absence of known risk factors such as arterial hypertension (HTN) or anticoagulation therapy (such as enoxaparin) as a COVID-19 thrombo-prophylaxis [14,15,16]. A prior study found that moderate-to-severe coronavirus patients benefit from anticoagulant therapy, such as enoxaparin, in terms of lowering mortality in this disease [17,18,19]. Then, we decided to detect the role of enoxaparin in decreasing the mortality rate in COVID-19 patients with pre-existing spontaneous ICH, even though it is a potential cause for worsening ICH.

Therefore, the objective of this study was to detect the outcomes of COVID-19 patients with pre-existing spontaneous intracerebral hemorrhage (comorbidity) and to detect the role of enoxaparin in decreasing the mortality rate in these cases, although it is a potential cause for worsening intracerebral hemorrhage.

## 2. Materials and Methods

### 2.1. The Study Design

A cohort study was conducted from May 2021 to December 2021, including 102 patients who were admitted to the neurosurgical unit with spontaneous ICH. The study protocol was approved by the Research Ethical Committee of Beni-Suef University (REC-H-PhBSU-20010, date of approval: March 2021) and was carried out at Beni-Suef University Hospital following the Helsinki Declaration. The patient’s family provided written informed consent. Of the total patients, 53 patients were infected with SARS-CoV-2, and they were laboratory-confirmed as having SARS-CoV-2 infection using reverse transcription-polymerase chain reaction (RT-PCR). The patients were treated for COVID-19 according to the guidelines of the World Health Organization (WHO) and the Egyptian protocol.

Through repeated negative RT-PCR tests performed within 45 days of the initial positive test, SARS-CoV-2 clearance was determined in all surviving patients.

#### 2.1.1. Inclusion Criteria

Patients were more than 18-years-old.Patients were admitted to the hospital for spontaneous intracerebral hemorrhage and acquired COVID-19 infection during the period of their hospital admission.Positive cases of COVID-19 by RT-PCR test.

#### 2.1.2. Exclusion Criteria

Severe hepatic disease patients.Patients who were pregnant or lactating.Patients aged less than 18-years-old.

### 2.2. Sampling Techniques

The data collected from patients who had pre-existing spontaneous intracerebral hemorrhage comorbidities after confirmation of their infection with COVID-19 were analyzed.

### 2.3. Data Collection

From the patients, the following was collected:
The participants’ characteristics and disease characteristics of COVID-19 disease symptoms and computed tomography (CT) scan.The neurological symptoms and CT brain findings on admission.Management of the studied patients regarding their COVID-19 disease and neurosurgical management.The outcomes of the patients who were studied.The association between mortality and various patient characteristics.The relation between enoxaparin and the outcome of the study.

Neurological parameters were collected for follow-up and monitoring of the patients to detect the link between patients’ neurological comorbidities and disease severity (recovery time) and clinical prognosis.

### 2.4. Statistical Analysis

The data were analyzed using IBM SPSS advanced statistics version 22 (SPSS Inc., Chicago, IL, USA), and the numerical data are expressed as mean and standard deviation or median and range, as appropriate. The qualitative data are expressed as frequency and percentage. The comparison between subgroups was performed using an independent *t*-test regarding scale variables, while the comparison regarding categorical variables was done using chi-squared or Fisher exact tests. A *p*-value < 0.05 was considered statistically significant.

## 3. Results

### 3.1. Participants’ Characteristics and COVID-19 Disease Characteristics (Symptoms and CT)

A total of 53 patients (30 males) completed the study. Their age (mean ± SD) was (59.1 ± 14.2). Regarding oxygen therapy, 60.4% of the patients required ordinary oxygen therapy, 18.9% required high-flow nasal oxygen therapy, and 20.8% required mechanical ventilation.

The characteristics of the studied patients and the disease characteristics of COVID-19 (symptoms and CT) are shown in Table 1.

### 3.2. The Neurological Symptoms and CT Brain Findings on Admission

The mean Glasgow Coma Scale (GCS) for the patients was 11.5. Of all the patients (regarding the conscious patients), 1.9% suffered from diminished vision, 5.7% suffered from headache, 35.8% had left-side weakness and 26.4% had a right-side weakness. Regarding the CT brain findings, 28.3% of the patients had midline shift and brain edema, 5.7% had bilateral basal ganglia hematoma, 7.5% had left basal ganglia hematoma, and 5.7% had right basal ganglia hematoma. 9.4% had bilateral thalamic hematoma, 5.7% had left thalamic hematoma, and 26.4% had a right thalamic hematoma. 3.8% had bilateral temporo-parietal hematoma, 9.4% had left temporo-parietal hematoma, and 1.9% had right temporo-parietal hematoma. 1.9% had a bilateral frontal hematoma and 1.9% had a right frontal hematoma. 24.6% had intraventricular hemorrhage (IVH), 3.8% had a pontine hemorrhage, and 5.7% had hydrocephalus. The mean hematoma size ± SD was 14.9 ± 11.9. The hematoma course on admission was stationary in 28.3% of the participants, regressive in 22.6%, and progressive in 49.1% of them. The neurological symptoms and CT brain findings are shown in Table 2.

### 3.3. Management of the Studied Patients regarding Their COVID-19 Disease and Neurosurgical Management

Of all the study patients, 32.1% received a prophylactic dose of enoxaparin on day 3, 52.8% received a prophylactic dose on day 3, followed by a therapeutic dose on day 7, and 15.1% did not receive enoxaparin. Regarding the neurosurgical management, 9.4% of the patients were evacuated, and 5.7% underwent ventriculo-subglia drain (VSD) surgery. Management of the studied patients regarding their COVID-19 disease and neurosurgical management is shown in Table 3.

### 3.4. The Outcomes of the Studied Patients

The condition of the chest and GCS had improved in 67.9% of the participants by the end of the study. The hematoma course increased in 49.1% of the patients. The midline shift, brain edema, and COVID symptoms improved in 67.9% of the participants. The outcomes of the patients who were studied are shown in Table 4.

### 3.5. The Relation between Mortality and Different Patients’ Characteristics

There was no significant difference in mortality between patients depending on gender and age. However, there was a significant difference in mortality among patients depending on chest condition, COVID-19 symptoms, GCS, midline shift, and brain edema (*p*-value < 0.001). There was a significant difference in mortality among patients with temporo-parietal hematoma (*p*-value = 0.047). There was a significant difference in mortality among patients depending on hematoma size (*p*-value < 0.001). The hematoma sizes (mean ± SD) were (4.612 ± 567) and (13.00 ± 3.202) for the living and dead patients, respectively. There was a significant difference in the mortality among patients depending on the treatment with enoxaparin, where 75% of the patients who did not receive enoxaparin died, while in patients who were administered enoxaparin, it was 24.4% (*p*-value = 0.026). The ICU stay duration (mean ± SD) was (12.28 ± 7.7) and (7.29 ± 5.193) for the living and dead patients, respectively (*p*-value = 0.019). The relationship between mortality and different patients’ characteristics is shown in Table 5.

### 3.6. The Relationship between Enoxaparin and the Outcome of the Study

Of all the patients who were treated with enoxaparin, 94.4% showed significant improvement in their chest condition, the mean Glasgow Coma Scale (GCS), the midline shift, brain edema, and COVID-19 symptoms (*p*-value = 0.01). An amount of 55.6% of the patients who were administered enoxaparin showed decreases in hematoma course and 25% who did not receive enoxaparin. Enoxaparin was given to 94.4% of the patients who were still alive. The relationship between enoxaparin and the study outcome is shown in Table 6.

## 4. Discussion

The findings of this study imply that coronavirus may worsen neurological disorders and, subsequently, mortality in people who have a pre-existing spontaneous intracerebral hemorrhage. This was also reported in previous studies [7,20,21].

According to a recent retrospective study, inpatients in a neurological unit who were infected with COVID-19 had a worse prognosis, developed severe cases, and experienced poorer outcomes than those who were not infected [13].

This is the first Egyptian study that shows how COVID-19 can make neurological symptoms worse in people who already have pre-existing spontaneous intracerebral hemorrhage and how many people die from it. 

In particular, approximately 32% of patients with pre-existing spontaneous intracerebral hemorrhage died after being infected with COVID-19 [22]. The patients who died had deteriorating neurological symptoms such as decreased GCS, increased hematoma course, med line shift, and brain edema, whereas COVID-19 infection causes the production of the central nervous system and systemic cytokines, prostaglandins, and monokines, which cause patients with neurological disorders to experience an abrupt decrease in cognition and a worsening prognosis [7,23,24].

In this study, about 53% of the dead patients required mechanical ventilation, indicating poor clinical outcomes with an elevated occurrence of severe conditions which led to mortality [25]. Worsening chest conditions were reported in all the dead patients.

In this study, the majority of hematoma patients had severe coronavirus. This could be because these people have risk factors for severe coronaviruses, such as age, hypertension, and diabetes. These risk factors also increase the mortality rate [26].

The majority of the dead patients in this study showed an increased size of hemorrhage and edema with midline shift, which caused compression on the brain stem and development of the hematoma course into a state of being progressive, leading to death. This may explain why most of the patients with progressive hematoma courses in this study died. These findings were confirmed by the study of TM Tu et al. [27].

For patients with increased hematoma course (hematoma size greater than 30 ml), the patients underwent surgery to evacuate the hematoma if the GCS score was between 7 and 13 and if the chest condition did not develop into severe pneumonia [28]. Of all the participants in this study, five of them underwent evacuation of the hematoma.

Ventricular subglia drain (VSD) is surgery to drain excess CSF from the ventricles (hydrocephalus) into the subglia space to relieve tension regardless of the score of GCS and the chest condition [29]. Three patients in this study suffered from hydrocephalus and urgently underwent VSD surgery.

For patients with a GCS score of less than 7 or whose chest condition has developed to a severe degree, dehydration therapy is used to monitor the cerebral edema and decrease the intracranial hemorrhage until their condition allows them to undergo surgery [30]. Mannitol, albumin, and glycerin fructose are the most commonly used osmotherapy medicines. Mannitol is the most commonly used dehydrant [31]. In this study, furosemide and mannitol were used as dehydration therapies. This combination results in a greater decrease in the amount of water in the brain than mannitol alone did. Furosemide enhances the effect of mannitol on plasma osmolality, causing a greater decrease in brain water content [32].

Difficulties with spontaneous breathing and respiratory secretion cleaning may exacerbate pneumonia in patients with pre-existing spontaneous intracerebral hemorrhage [33,34]. About 32.1% of the patients used a prophylactic dose (20 mg/0.4 ml twice daily) of enoxaparin on day 3 of their symptoms, and 52.8% used a prophylactic dose on day 3, followed by a therapeutic dose (60 mg/0.4 ml twice daily) on day 7. Anticoagulant use should be continued for four weeks for antithrombotic prophylaxis [35,36]. Approximately two-thirds of the cases included in this study did not progress to severe, which is especially intriguing given their previously severe and debilitating clinical profile. This unexpected outcome could be explained by the fact that these patients were given an anticoagulant for the prevention of thromboembolism, which is a result of commonly suffering from immobility for an extended duration. In reality, because of their severe neurological dysfunction, the majority of the patients in our study were bedridden, and anticoagulant medication was initiated early in their hospitalization, before SARS-CoV-2 infection. Early anticoagulant therapy with enoxaparin has been proposed as a helpful treatment since it has been linked to lower mortality in severe instances [37,38]. The anticoagulant effects could be explained through two-pronged mechanisms: anticoagulant, which minimizes the disease’s damaging effect, and anti-inflammatory, which prevents severe SARS-CoV-2 infection manifestations. The discovery that anticoagulants decrease cytokine release in a variety of inflammatory situations lends support to our interpretation [39]. Several studies have also found that enoxaparin improves coagulation issues in coronavirus patients and has anti-inflammatory properties that lower IL-6 and increase lymphocyte percentage. Enoxaparin appears to be useful in the treatment of coronavirus [40,41]. Furthermore, the antiviral activity of several anticoagulants, such as enoxaparin, has been predicted and supported by recent research [40,41]. However, the risk of bleeding problems from anticoagulant drugs in people with SARS-CoV-2 should not be ignored [41].

In terms of oxygen therapy, 60.4% of the patients in this study received standard oxygen therapy, 18.9% required high-flow nasal oxygen therapy, and 20.8% required mechanical ventilation. This shows that these patients developed moderate-to-severe chest conditions and required ICU care. Regarding the CT chest findings, 50.9% of the patients in this study were CORAD 3, 30.2% were CORAD 4, and 18.9% were CORAD 5. This means that people who have a pre-existing spontaneous intracerebral hemorrhage can quickly go from mild to severe COVID-19, which can lead to death.

In this study, there was no statistically significant difference in mortality among patients based on gender or age. However, there was a statistically significant difference in mortality among patients based on the comorbid pre-existing spontaneous intracerebral hemorrhage such as GCS, midline shift, brain edema, and hematoma course.

The hematoma sizes (mean ± SD) were (4.612 ± 2.567) and (13.00 ± 3.202) for the living and dead patients, respectively. The rising mortality rate could be explained by a hyperimmune response caused by cytokine storms, or by a direct viral invasion of human brain cells via transcribrial, hematogenous, and neural retrograde dissemination routes [42]. Furthermore, angiotensin-converting enzyme 2 (ACE2) receptors produced by capillary endothelial cells in the brain may be involved in SARS-CoV-2-induced neurological problems. Endothelial rupture in the brain causes irreparable brain injury, contributing to SARS-CoV-2 neurologic symptoms pathophysiology [42]. Moreover, higher D dimer and CRP levels caused by a state of high inflammation and activation of the coagulation cascade may result in cerebrovascular problems in coronavirus patients. As a result, the potential processes could include a combination of immunological, vascular, and neural variables. A neurologic manifestation of COVID-19 may occur in addition to preexisting intracerebral hemorrhage. A prior study found a link between COVID-19 and neurological illnesses, with more than 30% of hospitalized coronavirus patients developing neurological disorders [43]., ranging from mild to life-threatening in severity. As a result, it needs to be included in any individual’s differential diagnosis presenting with increasing neurological symptoms, particularly within the current epidemic. As a result, individuals with severe coronavirus infection, as well as previous neurological diseases, should be treated with special caution because they are more likely to die [44].

There was also a statistically significant difference in mortality between patients based on their treatment with enoxaparin (*p*-value = 0.026), where 75% of the patients who did not receive enoxaparin died. A recent study confirmed this finding, reporting that enoxaparin reduces mortality in coronavirus patients with moderate to severe cases [17]. Most patients (92.6%) who showed a decrease in hematoma course were administered enoxaparin. On the other hand, 76.9% of the patients who showed an increase in hematoma course were administered enoxaparin. As a result, we recommend using enoxaparin for patients with spontaneous intracranial hemorrhage and COVID-19 regardless of hematoma course because the rate of improvement was greater than the mortality rate after using enoxaparin in this study.

The increase in hematoma course in the patients who were administered enoxaparin, which caused their deaths, may have been due to large hematoma size and deterioration in GCS, as well as chest condition, which was also reported in previous studies [45,46]. Regarding the chest condition, oxygen saturation decreased in the blood, which led to anorexia and deteriorated consciousness and GCS. In patients with preexisting spontaneous intracerebral hemorrhage and COVID-19 infection, a lower GCS can be considered a predictor of poor outcomes [47,48].

## 5. Conclusions

Our findings suggest that more than half of the patients who were admitted to the neurosurgical unit with spontaneous intracerebral hemorrhage acquired COVID-19 infection during their hospitalization period. Nearly two-thirds of the cases included in this study did not progress to severe cases. A third of the study patients died as a result of deteriorating neurological and COVID-19 symptoms such as decreased GCS, increased hematoma course, med line shift, brain edema, and chest condition. The antithrombotic efficacy of enoxaparin given earlier, both before and throughout the COVID-19 infection, can considerably reduce mortality in COVID-19 individuals with pre-existing spontaneous intracerebral hemorrhage. It is recommended to use enoxaparin for patients with preexisting spontaneous intracerebral hemorrhage and COVID-19 regardless of hematoma size because, in this study, the rate of improvement was greater than the mortality rate after using enoxaparin.

## Figures and Tables

**Table 1 jpm-12-01822-t001:** Characteristics of the studied patients and disease characteristics of COVID-19 (symptoms and CT).

Characteristics	Values No. = 53 (%)
Age (mean ± SD)	59.1 ± 14.2
Sex Females Males	23 (43.4%) 30 (56.6%)
Fever	15 (28.3%)
Dyspnea	13 (24.5%)
Diarrhea	14 (26.4%)
nausea and vomiting	14 (26.4%)
need to oxygen Ordinary oxygen therapy high flow nasal Mechanical Ventilation	32 (60.4%) 10 (18.9%) 11 (20.8%)
CT chest CORAD 3 CORAD 4 CORAD 5	27 (50.9%) 16 (30.2%) 10 (18.9%)

**Table 2 jpm-12-01822-t002:** Neurological symptoms and CT brain findings on admission.

Items	Values No. = 53 (%)
GCS (mean ± SD) median(IQR)	11.5 ± 3.5 13 (8)
Headache	3 (5.7%)
diminished vision	1 (1.9%)
Weakness left side right side	19 (35.8%) 14 (26.4%)
CT findings -basal ganglia hematoma bilateral left right -thalamic hematoma bilateral left right -temporo-parietal hematoma bilateral left right -frontal hematoma bilateral right -midline shift and brain edema	3 (5.7%) 4 (7.5%) 3 (5.7%) 5 (9.4%) 3 (5.7%) 14 (26.4%) 2 (3.8%) 5 (9.4%) 1 (1.9%) 1 (1.9%) 1 (1.9%) 15 (28.3%)
IV hemorrhage	13 (24.6%)
pontine hemorrhage	2 (3.8%)
Hydrocephalus	3 (5.7%)
hematoma size(mean ± SD) median(IQR)	14.9 ± 11.9 9 (14)
Hematoma course on admission stationary regressive progressive	15 (28.3%) 12 (22.6%) 26 (49.1%)

**Table 3 jpm-12-01822-t003:** Management of the studied patients regarding their COVID-19 disease and neurosurgical management.

Items	Values No. = 53 (%)
Enoxaparin No Prophylactic dose on day 3 Prophylactic dose on day 3 then therapeutic dose on day 7	8 (15.1%) 17 (32.1%) 28 (52.8%)
surgical management Evacuation VSD(ventriculo-subglia drain)	5 (9.4%) 3 (5.7%)

**Table 4 jpm-12-01822-t004:** The outcomes of the studied patients.

Items	Values No. = 53 (%)
Chest Condition Improved Not improved	36 (67.9%) 17 (32.1%)
Headache Improved Not improved	3 (100%) 0 0
GCS Improved Not improved	36 (67.9%) 17 (32.1%)
Weakness Improved Not improved	0 0 33 (100%)
Hematoma course decreased increased	27 (50.9%) 26 (49.1%)
Medline shift and brain edema Improved Not improved	36 (67.9%) 17 (32.1%)
COVID-19 symptoms Improved Not improved	36 (67.9%) 17 (32.1%)

**Table 5 jpm-12-01822-t005:** Relations between mortality and different patients’ characteristics.

Characteristics	Alive (No. = 36)	Died (No. = 17)	*p*-Value
Age (mean ± SD)	59.14 ± 13.508	59.00 ± 16.109	0.974
Sex Females Males	17 (73.9%) 19 (63.3%)	6 (26.1%) 11 (36.7%)	0.413
need to oxygen Ordinary oxygen therapy high flow nasal Mechanical Ventilation	26 (81.3%) 8 (80.0%) 2 (18.2%)	6(18.8%) 2(20.0%) 9(81.8%)	
CT chest CORAD 3 CORAD 4 CORAD 5	20 (74.1%) 15 (93.8%) 1 (10.0%)	7 (25.9%) 1 (6.3%) 9 (90.0%)	
GCS (mean ± SD)	13.3 ± 1.7	7.53 ± 4.064	<0.001 *
midline shift and brain edema No yes	32 (84.2%) 4 (26.7%)	6 (15.8%) 11 (73.3%)	<0.001 *
basal ganglia hematoma No bilateral left right	29 (67.4%) 2 (66.7%) 4 (100.0%) 1 (33.3%)	14 (32.6%) 1 (33.3%) 0 (0.0%) 2 (66.7%)	FET 0.308
thalamic hematoma No bilateral left right	19 (61.3%) 3 (60.0%) 3 (100.0%) 11 (78.6%)	12 (38.7%) 2 (40.0%) 0 (0.0%) 3 (21.4%)	FET 0.520
tempro-parietal hematoma No bilateral left right	33 (73.3%) 1 (50.0%) 1 (20.0%) 1 (100.0%)	12 (26.7%) 1 (50.0%) 4 (80.0%) 0 (0.0%)	FET 0.047 *
frontal hematoma No bilateral right	34 (66.7%) 1 (100.0%) 1 (100.0%)	17 (33.3%) 0 (0.0%) 0 (0.0%)	FET 0.475
IV hemorrhage No Yes	26 (65%) 10 (76.9%)	14 (35%) 3 (23.1%)	FET 0.471
pontine hemorrhage No Yes	34 (66.7%) 2 (100.0%)	17 (33.3%) 0 (0.0%)	0.457 FET
Hydrocephalus No Yes	33 (66.0%) 3 (100.0%)	17 (34.0%) 0 (0.0%)	0.305
hematoma size(mean ± SD)	4.61 ± 2.567	13.00 ± 3.202	<0.001 *
Hematoma course on admission stationary regressive progressive	15 (100%) 12 (100%) 9 (34.6%)	0 (0%) 0 (0%) 17 (65.4%)	<0.001 * <0.001 * 0.333
Enoxaparin No Prophylactic dose on day3 Prophylactic dose on day3 then therapeutic dose on day 7	2 (25.0%) 13 (76.5%) 21 (75.0%)	6 (75.0%) 4 (23.5%) 7 (25.0%)	0.026 *
Chest Condition Improved Not improved	36 (100.0%) 0 (0.0%)	0 (0.0%) 17 (100.0%)	<0.001 *
Headache Improved Not improved	3 (100.0%) 0 (0.0%)	0 (0.0%) 0 (0.0%)	<0.001 *
GCS Improved Not improved	36 (100.0%) 0 (0.0%)	0 (0.0%) 17 (100.0%)	<0.001 *
Weakness (no = 39) Improved Not improved	0 (0%) 30 (91%)	0 (0%) 3 (9%)	<0.001 *
Hematoma size -Stable or decrease -Increase	36 (100.0%) 0 (0.0%)	0 (0.0%) 17 (100.0%)	<0.001 *
Medline shift and brain edema Improved Not improved	36 (100.0%) 0 (0.0%)	0 (0.0%) 17 (100.0%)	<0.001 *
COVID-19 symptoms Improved Not improved	36 (100.0%) 0 (0.0%)	0 (0.0%) 17 (100.0%)	<0.001 *
ICU stay (days)	12.28 ± 7.700	7.29 ± 5.193	0.019 *

(*) means significance.

**Table 6 jpm-12-01822-t006:** Relations between enoxaparin and the outcome of the study (presentations, mortality, and ICU stay).

Items	Not Administered Enoxaparin (No. = 8)	Administered Enoxaparin (No. = 45)	*p*-Value
Chest Condition Improved Not improved	2 (5.6%) 6 (35.3%)	34 (94.4%) 11 (64.7%)	0.010 *
GCS Improved Not improved	2 (5.6%) 6 (35.3%)	34 (94.4%) 11 (64.7%)	0.010 *
Weakness Improved Not improved	0 (0%) 2 (6%)	0 (0%) 31 (94%)	0.850
Hematoma course decreased increased	2 (7.4%) 6 (23.1%)	25 (92.6%) 20 (76.9%)	0.010 *
Medline shift and brain edema Improved Not improved	2 (5.6%) 6 (35.3%)	34 (94.4%) 11 (64.7%)	0.010 *
COVID-19 symptoms Improved Not improved	2 (5.6%) 6 (35.3%)	34 (94.4%) 11 (64.7%)	0.010 *
Mortality Alive Died	2 (5.6%) 6 (35.3%)	34 (94.4%) 11 (64.7%)	0.010 *
ICU stay (days)	2.38 ± 744	12.16 ± 6.974	0.019 *

(*) means significance.

## Data Availability

Data is available upon request.

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
