# Peer review of "The Outcomes of COVID-19 Patients with Spontaneous Intracerebral Hemorrhage Comorbidity and the Efficacy of Enoxaparin in Decreasing the Mortality Rate in Them: Single Egyptian Center Report"

_jpm, 2022, doi:10.3390/jpm12111822_

Round 1

Reviewer 1 Report

Dear Authors,

Congratulations on your paper, but some changes are required.

  1. Extensive editing of the English language and style is required.
  2. Line 88 „than18". Line 92 „COVID-19by"

  1. Your study contains some interesting data, but there are some points that are not clear and should be improved.

  1. In the first section, you describe the study population. A table of patient characteristics would help to get a better overview of the patient population. 

  1. Please provide some images of the cases' CT scans. Also, you should detail one or two cases.

  1. The tables can be optimized as a design.

  1. A comparison between your cases and resembling cases from the literature is needed (you should make a table to be easy to read).

Author Response

  1. Extensive editing of the English language and style is required.

Author: I have corrected the grammatical errors in the manuscript by (grammarly) program, Thank you very much.

  1. Line 88 „than18". Line 92 „COVID-19by"

Author: Done, thank you

  1. Your study contains some interesting data, but there are some points that are not clear and should be improved.

Author: improved as much as possible, thank you

  1. In the first section, you describe the study population. A table of patient characteristics would help to get a better overview of the patient population. 

Author: The study population was shown in table 1 which titled the characteristics of the studied patients, thank you 

  1. Please provide some images of the cases' CT scans. Also, you should detail one or two cases.
  1. The tables can be optimized as a design.

 Author: improved as much as possible, thank you

  1. A comparison between your cases and resembling cases from the literature is needed (you should make a table to be easy to read).

Author: we compared our cases findings with other studies in discussion, thank you

Reviewer 2 Report

This center report shows different outcomes, based on CT findings and clinical courses, in a population with spontaneous intracerebral hemorrhage infected with COVID19, based on antithrombotic therapy administration. Authors suggest a possible role of enoxaparin in decreasing the mortality in this population, although it is a potential cause for worsening ICH.

I think that all the significative relations identified in the study are affected by the inadequate number of the sample of the two groups. Moreover, a second issue is the lack of patient stratification based on early brain lesions severity. Please provide this information.

Introduction: CO-RAD scale should be explained in introduction and Tab 1

Line 76: How was randomization carried out? 

Line 136: The maximum GCS score is 15; mean (11.5) plus SD (3.9) is 15.4

Line 153-155: The two groups are not homogeneous, 8 untreated vs 45 untreated. 

Tab 3 does not provide additional information; it could be deleted

Author Response

I think that all the significative relations identified in the study are affected by the inadequate number of the sample of the two groups. Moreover, a second issue is the lack of patient stratification based on early brain lesions severity. Please provide this information.

Author: During the period from May 2021 to December 2021, only 102 patients were admitted to the neurosurgical unit with spontaneous ICH, as a result of that, the sample size is not small at the time of study.

Regarding the early brain lesions severities for the participants were shown in table 2 (for ex. Hematoma course on admission)

Introduction: CO-RAD scale should be explained in introduction and Tab 1

Author: added, thank you

Line 76: How was randomization carried out? 

Author: corrected, thank you

Line 136: The maximum GCS score is 15; mean (11.5) plus SD (3.9) is 15.4

Author: corrected, thank you

Line 153-155: The two groups are not homogeneous, 8 untreated vs 45 untreated.           

Author: it is a cohort study aimed to evaluate the results of patients management who were admitted to the neurosurgical unit with spontaneous ICH, as a result of that, we reported in this study how they were managed in Beni-Suef university hospital and extracted our results regardless group homogeneity.

Tab 3 does not provide additional information; it could be deleted

Author: removed, thank you

Round 2

Reviewer 2 Report

Thank you for the clarifications